# Association of *TP53* Alteration with Tissue Specificity and Patient Outcome of *IDH1*-Mutant Glioma

**DOI:** 10.3390/cells10082116

**Published:** 2021-08-17

**Authors:** Balazs Murnyak, L. Eric Huang

**Affiliations:** 1Department of Neurosurgery, Clinical Neurosciences Center, University of Utah, Salt Lake City, UT 84132, USA; balazs.murnyak@utah.edu; 2Department of Oncological Sciences, Huntsman Cancer Institute, University of Utah, Salt Lake City, UT 84112, USA

**Keywords:** glioma, IDH, isocitrate dehydrogenase, pan-cancer, patient outcome, survival, tissue specificity, *TP53*, tumor suppressor

## Abstract

Since the initial discovery of recurrent isocitrate dehydrogenase 1 (*IDH1)* mutations at Arg132 in glioma, *IDH1* hotspot mutations have been identified in cholangiocarcinoma, chondrosarcoma, leukemia, and various other types of cancer of sporadic incidence. Studies in glioma and leukemia have helped promote the theory that *IDH1* mutations are an oncogenic event that drives tumorigenesis in general. Through bioinformatic analysis of more than 45,000 human pan-cancer samples from three independent datasets, we show here that *IDH1* mutations are rare events in human cancer but are exclusively prevalent in WHO grade II and grade III (lower-grade) glioma. Interestingly, alterations in the tumor-suppressor gene *TP53* (tumor protein p53) co-occur significantly with *IDH1* mutations and show a tendency of exclusivity to *IDH2* mutations. The co-occurrence of *IDH1* mutation and *TP53* alteration is widespread in glioma, particularly in those harboring *IDH1^R132H^*, *IDH1^R132G^*, and *IDH1^R132S^*, whereas co-occurrence of *IDH1^R132C^* and *TP53* alteration can be found sporadically in other cancer types. In keeping with the importance of p53 in tumor suppression, *TP53* status is an independent predictor of overall survival irrespective of histological and molecular subgroups in lower-grade glioma. Together, these results indicate tissue specificity of *IDH1* hotspot mutation and *TP53* alteration and the importance of *TP53* status as a predictor of patient outcome in lower-grade glioma.

## 1. Introduction

The *IDH1* gene encodes NADP^+^-dependent IDH localized in the cytoplasm and peroxisomes [1,2,3,4,5]. This enzyme not only catalyzes the oxidative decarboxylation of isocitrate to 2-oxoglutarate but also is critical to reductive carboxylation, which is required for lipogenesis in hypoxia and redox homeostasis during anchorage-independent growth [6,7]. In agreement with its physiological function of regulating the intracellular NADP^+^/NADPH ratio [8], IDH1 plays an important role in metabolic adaption in physiology and cancer biology.

Earlier studies revealed widespread *IDH1* mutations at Arg132, an active site of the enzyme, in WHO grade II and III (referred to as lower-grade) glioma and in WHO grade IV secondary glioblastoma [9,10,11]. Among these hotspot mutations, the *IDH1^R132H^* frequency was >90%, whereas non-canonical *IDH1* mutations, including *IDH1^R132C^*, *IDH1^R132G^*, *IDH1^R132S^*, and *IDH1^R132L^* (referred to collectively as *IDH1^R132X^*), were at very low frequencies [12]. Furthermore, mutations in the *IDH2* gene (encoding a mitochondrial NADP^+^-dependent enzyme) at the analogous Arg172 were relatively uncommon in lower-grade glioma and non-existent in glioblastoma [13]. Although they were thought to be virtually exclusive in glioma [11,14], further mutational analyses revealed *IDH1* and *IDH2* mutations, as well as *IDH2* Arg140 mutations, in various cancer types such as myeloid neoplasia, chondrosarcoma, cholangiocarcinoma, and prostate cancer [2,15].

*IDH1* and *IDH2* mutations acquire a neomorphic function that produces (D)-2-hydroxyglutarate (D-2HG) from the NADPH-dependent reduction of 2-oxoglutarate [16]. High levels of D-2HG induce histone and DNA hypermethylation through competitive inhibition of histone and DNA demethylases, thereby blocking cell differentiation and promoting neural stem-like phenotype [17,18,19]; however, neural stem-cell marker genes such as *NES* and *PROM1* are generally downregulated in IDH-mutant glioma [20]. The finding that *IDH1* and *IDH2* mutations occur in various other cancer types has spurred further interest in cancer metabolism, epigenetic regulation, and therapeutic targeting, also promoting the idea that these mutations drive tumorigenesis in general [1,2,3,4,5,21], despite how *IDH1* and *IDH2* mutations promote gliomagenesis remains unclear [13,22].

Genetically, lower-grade gliomas with *IDH1* and *IDH2* mutations are associated with either *TP53* and/or *ATRX* alteration or 1p/19q codeletion [23,24]. Although *TP53* is among the most-mutated tumor-suppressor genes in human cancer [25,26,27], the biological significance of *TP53* alteration in lower-grade glioma requires further investigation. In this study, we analyzed three independent pan-cancer datasets and revealed that the association of *IDH1* mutation with *TP53* alteration is specific to glioma, which indicates a tissue-specific role for *TP53* alteration in gliomagenesis.

## 2. Materials and Methods

### 2.1. Pan-Cancer Datasets

Three independent pan-cancer datasets: TCGA PanCancer dataset (TCGA_PanCancer); MSK-Impact pan-cancer dataset (MSK_Impact); and a combined, non-redundant pan-cancer dataset (Non-Redundant), were downloaded from cBioPortal [28,29]. Downloaded data included study ID, sample ID, patient ID, and patient status and survival, with matched genetic alteration data of *IDH1*, *IDH2*, *TP53*, *CDKN2A*, *CDKN2B*, *CIC*, *FUBP1*, and 1p/19q codeletion.

TCGA_PanCancer consists of 32 studies comprising 10,953 patients/10,967 samples from cancer types including bladder urothelial carcinoma (BLCA, *n* = 411), cholangiocarcinoma (CHOL, *n* = 36), colorectal adenocarcinoma (COADREAD, *n* = 594), breast invasive carcinoma (BRCA, *n* = 1084), brain lower-grade glioma (LGG, *n* = 514), glioblastoma (GBM, *n* = 592), esophageal adenocarcinoma (ESCA, *n* = 182), stomach adenocarcinoma (STAD, *n* = 440), head and neck squamous cell carcinoma (HNSC, *n* = 523), liver hepatocellular carcinoma (LIHC, *n* = 372), lung adenocarcinoma (LUAD, *n* = 566), lung squamous cell carcinoma (LUSC, *n* = 487), acute myeloid leukemia (LAML, *n* = 200), ovarian serous cystadenocarcinoma (OV, *n* = 585), pancreatic adenocarcinoma (PAAD, *n* = 184), skin cutaneous melanoma (SKCM, *n* = 448), sarcoma (SARC, *n* = 255), and uterine corpus endometrial carcinoma (UCEC, *n* = 529).

MSK_Impact consists of 10,336 patients/10,945 profiled samples, including glioma (*n* = 553), hepatobiliary cancer (Hepatobiliary, *n* = 355), bone cancer (Bone, *n* = 135), skin cancer, non-melanoma (SKNM, *n* = 148), small cell lung cancer (SCLC, *n* = 82), melanoma (*n* = 365), mature T and NK neoplasms (Mature T and NK, *n* = 134), uterine sarcoma (USARC, *n* = 93), small bowel cancer (Small Bowel, *n* = 35), and central nervous system cancer (CNS, *n* = 48).

Non-Redundant consists of 152 studies published from numerous institutions comprising 25,016 patients/26,922 samples with various cancer types, including acute myeloid leukemia or myelodysplastic syndromes (mnm), skin cutaneous melanoma (skcm), metastatic melanoma (mel), uterine carcinoma (ucs), cutaneous squamous cell carcinoma (cscc), primary central nervous system lymphoma (pcnsl), esophageal squamous cell carcinoma (escc), esophageal adenocarcinoma (esca), ampullary carcinoma (ampca), and basal cell carcinoma (bcc).

### 2.2. Data Analysis

Relevant data were extracted and processed with GraphPad Prism version 9.0 software (GraphPad Software, San Diego, CA, USA) to present genetic alteration frequency and occurrence. Statistical significance in frequency difference was determined by paired *t*-test or Fisher’s exact test, as specified, with two-tailed *p*-values. Overall survival was analyzed by the Mantel-Cox log-rank test as previously described [30,31]. Multivariate Cox proportional hazards analysis was performed with SPSS Statistics (IBM) software by including *TP53* status, *IDH1* status, age, sex, and histological type, as previously described [31].

## 3. Results

### 3.1. IDH1 Hotspot Mutations Are a Rare Event but Prevalent Exclusively in Lower-Grade Glioma

To obtain a landscape of *IDH1* and *IDH2* mutations in human cancer, we analyzed samples from three independent pan-cancer datasets: TCGA PanCancer dataset (TCGA_PanCancer); MSK-Impact pan-cancer dataset (MSK_Impact); and a combined, non-redundant pan-cancer dataset (Non-Redundant), which comprised a total of 45,228 human samples with various cancer types (see Materials and Methods).

Analysis of TCGA_PanCancer revealed overall frequencies of *IDH1* and *IDH2* alterations, including mutation, homozygous deletion, and amplification, at 6% and 2%, respectively. Among these alterations, *IDH1* hotspot mutations were <5% (or 480) and isolated alterations were <1%, whereas nearly 95% samples had no alteration (Figure 1; Table 1). Likewise, 99% of samples showed no alteration in *IDH2*, and only 0.4% (or 46) samples had Arg140 or Arg172 mutations (Appendix A). The low frequencies of *IDH1* and *IDH2* mutations in human cancer were confirmed with MSK_Impact; the overall frequencies of *IDH1* and *IDH2* alterations were 3% and <1%, respectively; and *IDH1* and *IDH2* hotspot mutations were 2% and 0.3% (or 260 and 31), respectively (Figure 1; Table 1 and Appendix A). Moreover, similar results were obtained from Non-Redundant, with *IDH1* and *IDH2* hotspot mutations at 1% and 0.4%, respectively (Figure 1; Table 1 and Appendix A). Thus, the overall frequencies of *IDH1* and *IDH2* hotspot mutations were 2% and 0.4%, respectively. In contrast to the high frequencies of *TP53* alteration averaging 32% (Figure 1), these results indicate that both *IDH1* and *IDH2* hotspot mutations are rare events in human cancer.

The frequency of *IDH1* mutation in lower-grade glioma, however, was conspicuously high (77%) in TCGA_PanCancer, with cholangiocarcinoma and acute myeloid leukemia much lower at 14% and <10%, respectively (Figure 2A). *IDH2* mutation was most common in acute myeloid leukemia (11%), followed by <6% in cholangiocarcinoma; however, *IDH2* amplification was more common, albeit at low frequencies, among various cancer types (Figure 2B). In MSK_Impact, *IDH1* alteration was 33% in various types of gliomas and 14% in hepatobiliary cancer (Figure 2D), whereas *IDH2* mutation was seen most frequently in mature T and NK neoplasms (Figure 2E). In Non-Redundant, the frequency of *IDH1* mutation was 100% in lower-grade glioma, and cancer types with *IDH2* mutation >10% included acute myeloid leukemia, myelodysplastic syndromes, and primary central nervous system lymphoma (Appendix A). Again, *IDH2* amplification was seen particularly in prostate adenocarcinoma, pancreatic adenocarcinoma, melanoma, and invasive breast carcinoma. In contrast, *TP53* mutation was widespread among various cancer types (Figure 2C,F and Appendix A). Therefore, although both *IDH1* and *IDH2* mutations are rare events in human cancer, the prevalence of *IDH1* mutation in lower-grade glioma suggests a tissue-specific role in tumorigenesis.

### 3.2. Co-Occurrence of IDH1 Hotspot Mutation and TP53 Alteration Predominantly in Glioma

Despite the low frequency of *IDH1* mutation in human cancer, further analysis revealed significant to extremely significant co-occurrence of *IDH1* and *TP53* alterations but mutual exclusivity between *IDH2* and *TP53* alterations in all three datasets (Table 2 and Table S2). Furthermore, the overall frequency of *IDH1* hotspot mutation co-occurring with *TP53* alteration was 49% versus 23% for the co-occurrence of *IDH1* isolated alteration and *TP53* alteration (Table 3). Specifically, the co-occurrence frequency remained above 50% for *IDH1^R132H^*, *IDH1^R132G^*, and *IDH1^R132S^*, but much lower for *IDH1^R132C^* and *IDH1^R132L^* (Table 4). Consistent with the mutual exclusivity, only 6% (11/182) of *IDH2* hotspot mutations co-occurred with *TP53* alteration (Appendix A). These results indicate that *TP53* alterations exhibit a tendency of co-occurring with *IDH1*, but not *IDH2*, mutations in human cancer.

To assess whether such co-occurrence is cancer-type specific, we extracted all cancer types harboring *IDH1* hotspot mutation and *TP53* alteration. Interestingly, 97% of the co-occurrences were in lower-grade glioma and glioblastoma in TCGA_PanCancer, with the rest including melanoma and lung adenocarcinoma (Figure 3A; Appendix A). In particular, *IDH1^R132H^*, *IDH1^R132G^*, and *IDH1^R132S^* co-occurrences were exclusive to glioma, whereas *IDH1^R132C^* co-occurrence was seen in various cancer types (Figure 3B; Appendix A). In MSK_Impact, 87% of the co-occurrences were gliomas of various types, and the rest included cholangiocarcinoma, lung adenocarcinoma, and chondrosarcoma (Figure 3C; Appendix A). Again, *IDH1^R132H^*, *IDH1^R132G^*, and *IDH1^R132S^* co-occurrences were virtually exclusive to glioma except for single cases of *IDH1^R132H^* astroblastoma, *IDH1^R132H^* adenoid cystic carcinoma, and *IDH1^R132G^* lung adenocarcinoma (Figure 3D; Appendix A). Lastly, in Non-Redundant, gliomas of various types accounted for 90% of the co-occurrences, whereas acute myeloid leukemias were only 4% (Figure 3E; Appendix A). Specifically, 96% of the *IDH1^R132H^* co-occurrences and 60% of the *IDH1^R132G^* co-occurrences were in glioma (Figure 3F; Appendix A). As expected, co-occurrences of *IDH2* hotspot mutations and *TP53* alterations were extremely rare; there were a total of 11 cases among all three datasets, including 4 cases of lower-grade glioma, 3 cases of acute myeloid leukemia, and 2 cases of basal cell carcinoma (Appendix A). Therefore, the virtual exclusivity of co-occurrence of *IDH1* hotspot mutation and *TP53* alteration in glioma indicates the importance of *TP53* alteration in *IDH1*-mutant gliomagenesis.

### 3.3. Differential Co-Occurrence Frequencies between IDH1^R132H^ and IDH1^R132X^ in Glioma

Non-canonical *IDH1^R132X^* occurs in 8% of lower-grade glioma harboring *IDH1* hotspot mutations [13]. In keeping with the notion that co-occurrence of *IDH1* hotspot mutation and *TP53* alteration is glioma-specific, the mean co-occurrence frequency was fivefold greater in glioma than in non-glioma (Figure 4A); however, the difference in *IDH1^R132H^* co-occurrence frequencies between glioma and non-glioma was not statistically significant (Figure 4B), even though *IDH1^R132H^* occurred in 92% in lower-grade glioma harboring *IDH1* hotspot mutations [13]. In contrast, whereas *IDH1^R132C^* is the major form in chondrosarcoma, cholangiocarcinoma, and acute myeloid leukemia [13], the co-occurrence of *IDH1^R132C^* and *TP53* alteration was nearly eightfold greater in glioma compared with non-glioma (Appendix A), as was the co-occurrence of combined *IDH1^R132X^* (Figure 4B).

The significant co-occurrence of *IDH1* hotspot mutation and *TP53* alteration in glioma was in accordance with the consistently high frequencies found across various histological subtypes, including glioblastoma, from both TCGA_PanCancer and MSK_Impact (Table 5). In contrast, the co-occurrence frequency of *IDH1^R132H^* and *TP53* alteration in oligodendroglioma averaged 17% versus >90% in astrocytoma and glioblastoma. Despite the rare occurrence of *TP53* alteration in oligodendroglioma [24], the *TP53* alteration frequency in *IDH1^R132X^* oligodendroglioma was 100% (5/5), significantly greater than that of *IDH1^R132H^* oligodendroglioma (*p* = 0.0012, Fisher’s exact test). Of note, given the mutual exclusivity of *TP53* alteration and 1p/19q codeletion in IDH-mutant lower-grade glioma [24], none of the *IDH1^R132X^* oligodendrogliomas harbored *CIC* and/or *FUBP1* mutations that are associated with 1p/19q codeletion.

### 3.4. TP53 Status Is an Independent Predictor of Patient Survival in Lower-Grade Glioma

Glioma patients with *IDH1* hotspot mutations are known to have better survival than those without such mutations [11,24,32], but astrocytoma patients with non-canonical *IDH1^R132X^* have even longer survival than those with *IDH1^R132H^* [33]. By following the latest cIMPACT-NOW recommendation that IDH-mutant gliomas harboring homozygous *CDKN2A/B* deletion are equivalent to *IDH1*-wildtype [34], we not only confirmed this finding in the TCGA-LGG dataset but, more importantly, observed the role of *TP53* status in patient survival (Figure 5A,B). *TP53* status distinguished survival in both *IDH1^R132H^* and *IDH1^R132X^* subgroups despite the significant increase in overall survival in patients with *IDH1^R132X^* compared with those with *IDH1^R132H^*. The clustering of *TP53*-wildtype *IDH1^R132H^* glioma and *TP53*-altered *IDH1^R132X^* glioma in overall survival underscored the paramount importance of *TP53* status in the outcomes of glioma patients. Moreover, similar significant associations were observed in the entire cohort and in histological and molecular subgroups including oligodendroglioma and *IDH1*-wildtype glioma (Figure 5C and Appendix A), in agreement with the tumor-suppressive function of p53 in human cancer [27,35,36].

To confirm these results, we performed a multivariate Cox proportional hazards analysis and found that *TP53* status was significant in the *IDH1* hotspot mutation subgroup (HR = 2.079; 95% CI: 1.083–3.992; *p* = 0.028), in the oligodendroglioma subgroup (HR = 2.001; 95% CI: 1.032–3.879; *p* = 0.040) (Table 6 and Table 7), as well as in the entire cohort (HR = 1.809; 95% CI: 1.327–3.150; *p* = 0.001) and the *IDH1*-wildtype subgroup (HR = 2.572; 95% CI: 1.378–4.802; *p* = 0.003) (Appendix A). Therefore, *TP53* status is an independent predictor of patient survival in lower-grade glioma irrespective of molecular and histological subclassifications.

## 4. Discussion

Through a survey of more than 45,000 pan-cancer samples, we observed that *IDH1* and *IDH2* hotspot mutations are uncommon (2%) and extremely rare (0.4%), respectively, in human cancer, a finding in agreement with an independent pan-cancer analysis [37]. Therefore, despite being prevalent in glioma, as reported previously [10,11,14,38], these mutations appear to be selected against in tumorigenesis, which is seemingly at odds with the general thought that these mutations induce oncogenic transformation through epigenetic and metabolic reprogramming resulting from high levels of D-2HG [13]. *IDH1^R132H^*, the most common form in glioma, produces the least amount of D-2HG and correlates with worse survival compared with the rare *IDH1^R132X^* and *IDH2*-R172 mutations, which produce higher levels of D-2HG [12,33,39]. Although the rare occurrence of these mutations in glioma has been attributable to the “cytotoxicity” of high levels of D-2HG [12,33], D-2HG sensitizes cells to ferroptosis [40]—an iron-dependent form of nonapoptotic cell death likely involved in tumor suppression [41]. D-2HG also exhibits tumor-suppressive activities through the inhibition of aerobic glycolysis in both IDH-mutant and IDH-wildtype leukemia cells [42]. Together with our previous studies showing that *IDH1* hotspot mutations are intrinsically tumor suppressive [30,43,44], these findings may provide an explanation for the rare occurrence of *IDH1* and *IDH2* hotspot mutations in human cancer.

The prevalence of *IDH1* hotspot mutation in glioma and its co-occurrence with *TP53* alteration indicate a tissue-specific role in gliomagenesis [13]. Tissue specificity in cancer is best evidenced by hereditary cancer predisposition syndromes in which the underlying gene defects, such as mutations in *APC*, *BRCA1*, and *VHL*, are associated with a high risk of developing tissue-specific cancer types [45]. In nonhereditary cancers, a subset of recurring genetic alterations can be identified to be associated with a particular type of cancer [45,46]. What drives tissue specificity in cancer, however, is complex even though numerous possibilities, including cell of origin, heterogeneity, epigenetic state, and environment, have been proposed [45,46]. In keeping with this, studies have shown the requirement of *Trp53* knockout/down to recapitulate a less aggressive phenotype of *IDH1^R132H^* glioma compared with *IDH1*-wildtype glioma [47,48,49]; however, the mechanism by which *TP53* alteration contributes to gliomagenesis remains unclear.

Interestingly, p53-mediated ferroptosis, a novel function of p53, has been implicated in tumor suppression independent of its previously recognized tumor-suppressive activities in cell cycle, apoptosis, and senescence [50,51,52,53,54]. Given the strongest display of ferroptosis-sensitive gene signature in IDH-mutant lower-grade glioma among all cancer types [55], we speculate that *TP53* alteration is required to inhibit ferroptosis for gliomagenesis, especially for *IDH1^R132X^* gliomas, including oligodendroglioma, that are supersensitive to ferroptosis owing to the higher levels of D-2HG. For *IDH1^R132H^* gliomas that are relatively less sensitive to ferroptosis, alternative tumor-suppressor pathways, such as 1p/19q codeletion, must be inactivated. Furthermore, our previous studies suggested the importance of the glutamate-rich cerebral environment in IDH-mutant lower-grade gliomagenesis [44,56], in agreement with the role of environment for tissue specificity in cancer. Therefore, the requirement of *TP53* alteration and a glutamate-rich environment in gliomagenesis warrants further investigation to account for the prevalence of *IDH1* hotspot mutations in glioma.

The tendency of mutual exclusivity between *IDH2* and *TP53* alteration in human cancer, including glioma, is intriguing, which may suggest alternative mechanisms of tumor-suppressor gene inactivation in tumorigenesis. Given the higher levels of D-2HG and its association with better survival [33,39], understanding how *IDH2*-R172 glioma cells overcome D-2HG induced sensitization to ferroptosis will shed light on the mechanism of *IDH2*-mutant gliomagenesis and the rare occurrence of such a mutation in human cancer.

Although the p53 tumor-suppressor pathway is altered at the frequency of 87% in glioblastoma [57], previously, the *TP53* status has not been associated with patient survival outcomes despite the well-established association of *TP53* alteration with IDH mutations in glioma [23,24,58,59]. Interestingly, *TP53* alteration has been associated with poor outcomes in pediatric H3 K27M-mutant glioma [60]. Likewise, in lower-grade glioma we provided evidence that *TP53* status is an independent predictor of overall survival in various molecular and histological subgroups, including the *IDH1^R132X^* subgroup, which was recently reported to have better outcomes than the *IDH1^R132H^* subgroup [33]. Interestingly, we observed similar overall survival between the *TP53*-wildtype *IDH1^R132H^* subgroup and the *TP53*-altered *IDH1^R132X^* subgroup based on the available size of samples; however, the paramount importance of *TP53* status in *IDH1^R132X^* glioma requires validation with independent cohorts. Similarly, the importance of *TP53* status in oligodendroglioma patient survival also requires validation. Moreover, additional genetic events, such as *ATRX*, *TERT*, and *BRAF*, which also frequently occur in IDH-mutant glioma [24,61,62], should be considered in future investigations.

## 5. Conclusions

*IDH1* hotspot mutations are rare events in human cancer but prevalent in glioma. The co-occurrence of *IDH1* hotspot mutation and *TP53* alteration indicates the tissue specificity of these genetic changes in gliomagenesis. *TP53* status is an important predictor of overall survival in lower-grade glioma.

## Figures and Tables

**Figure 1 cells-10-02116-f001:**
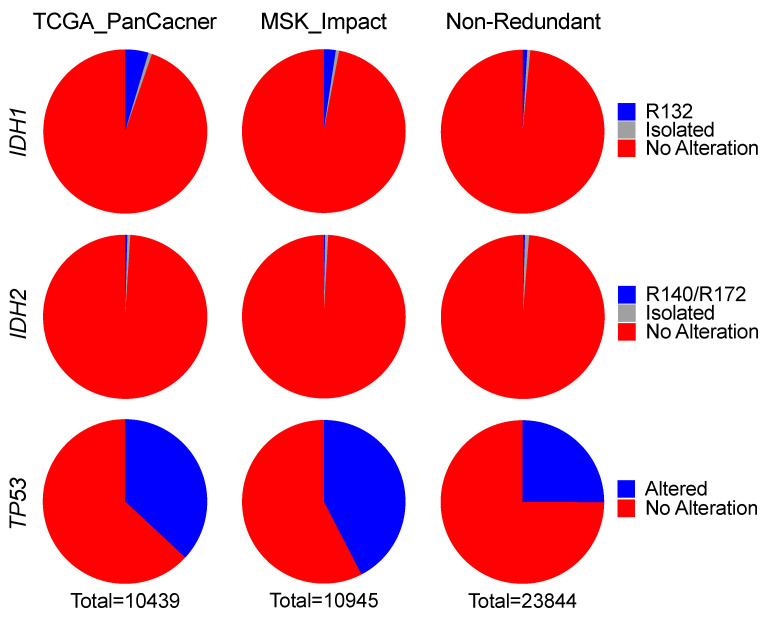
*IDH1* and *IDH2* alterations are rare in human cancer. Recurrent Arg132 mutation (R132) in *IDH1*, isolated genetic events (isolated), and no alteration were extracted from the TCGA_PanCancer, MSK_Impact, and Non-Redundant datasets. *IDH2* Arg140 and Arg172 (R140/R172) mutations and *TP53* alterations were analyzed similarly.

**Figure 2 cells-10-02116-f002:**
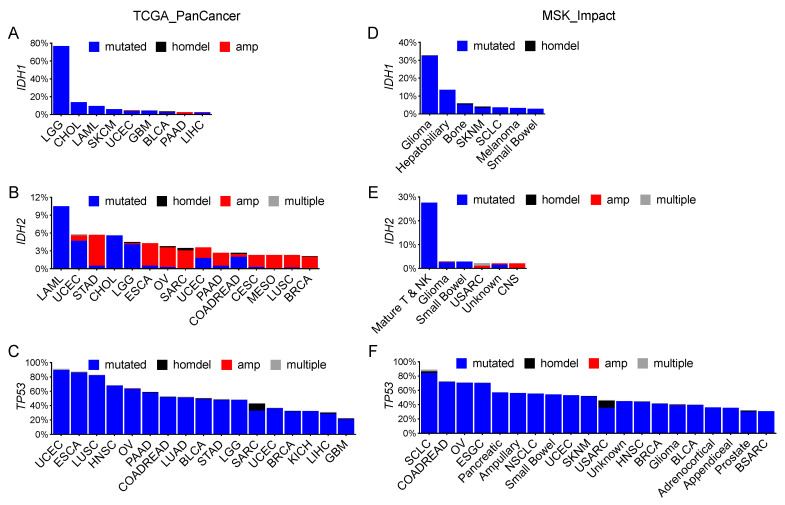
Distinctive distribution of *IDH1*, *IDH2*, and *TP53* alterations in human cancer. Extraction of *IDH1*, *IDH2*, and *TP53* alterations from specified datasets revealed a high frequency of *IDH1* mutations exclusively in glioma (**A**,**D**). Whereas relatively high frequencies of *IDH2* mutation were limited to hematopoietic neoplasms, *IDH2* amplification (amp) was seen in more cancer types (**B**,**E**) and *TP53* alteration was widespread (**C**,**F**). The cutoff is 2% for *IDH1* and *IDH2* and 20% (**C**) or 30% (**F**) for *TP53*.

**Figure 3 cells-10-02116-f003:**
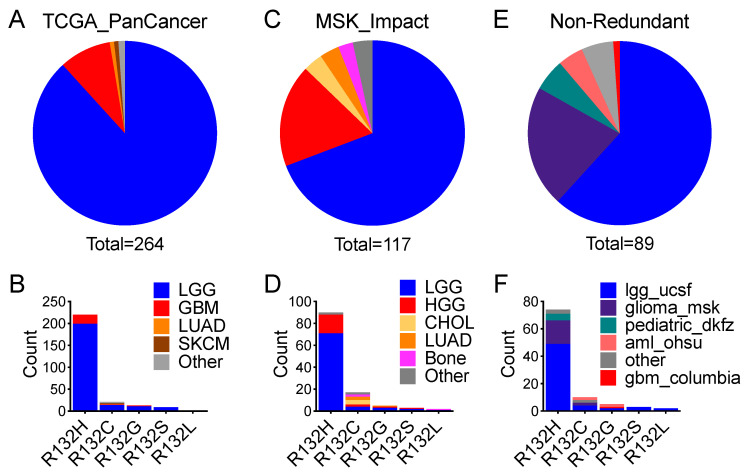
Co-occurrence of *IDH1* hotspot mutation and *TP53* alteration is predominantly in glioma. Analysis of TCGA_PanCancer (**A**,**B**), MSK_Impact (**C**,**D**), and Non-Redundant (**E**,**F**) datasets reveals co-occurrence of *IDH1* hotspot mutations and *TP53* alterations overwhelmingly in glioma and rarely in other cancer types, as presented in pie charts (top) and column charts (bottom) where sample counts of the cancer types are in reference to specific types of *IDH1* mutation. Of note, the cancer types in pediatric_dkfz (**F**) are high-grade glioma.

**Figure 4 cells-10-02116-f004:**
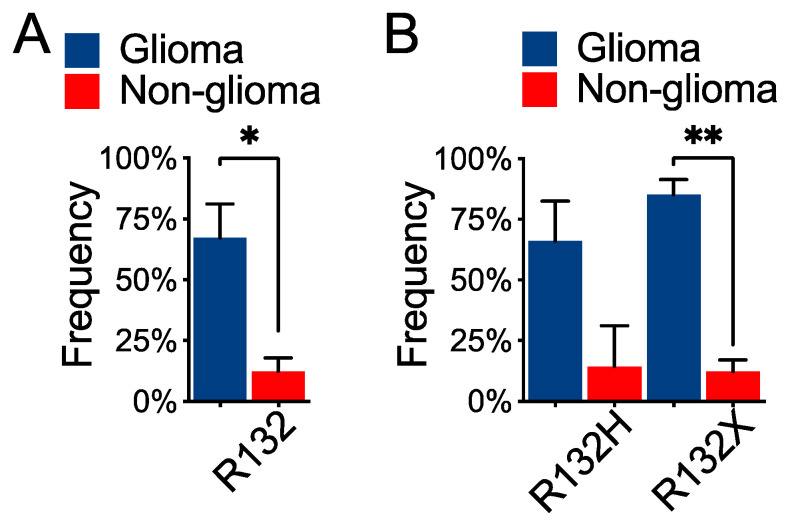
Higher frequencies of co-occurrence of *IDH1* hotspot mutation and *TP53* alteration in glioma. Glioma and non-glioma were compared for their co-occurrence frequencies of *TP53* alteration and *IDH1*-R132 mutation (**A**), and *TP53* alteration and *IDH1^R132H^* or *IDH1^R132X^* (**B**). * *p* < 0.05; ** *p* < 0.01.

**Figure 5 cells-10-02116-f005:**
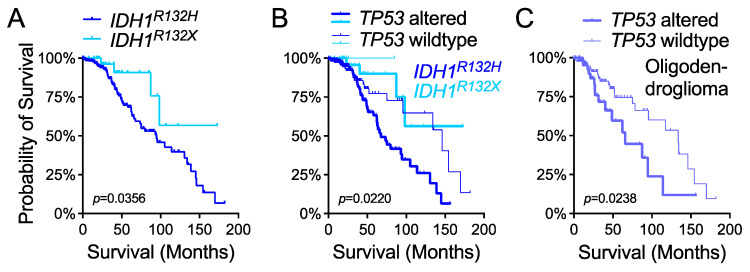
*TP53* status determines patient survival outcomes in molecular and histological subgroups of lower-grade glioma in TCGA_PanCancer. Differences in overall survival were analyzed between *IDH1^R132H^* and *IDH1^R132X^* subgroups (**A**) and among those of *TP53* wildtype (thin line) and *TP53* altered (thick line) (**B**). Overall survival was also analyzed in the oligodendroglioma subgroup with respect to *TP53* status (**C**). Two-tailed *p*-values are specified.

**Table 1 cells-10-02116-t001:** Occurrence and frequency of *IDH1* alteration in human cancer.

Dataset	Samples	R132		Isolated		No Alteration	
TCGA_PanCancer	10,439	480	5%	74	1%	9884	95%
MSK_Impact	10,945	260	2%	74	1%	10,610	97%
Non-Redundant	23,844	219	1%	148	1%	23,256	98%
Total	45,228	961	2%	296	1%	43,750	97%

**Table 2 cells-10-02116-t002:** Co-occurrence of *IDH1* and *TP53* alterations in human cancer.

Dataset	*IDH1*	*TP53*	Both	Neither	Log2 OR	*p*-Value	*q*-Value	Tendency
TCGA_PanCancer	256	3557	299	6327	1.055	<0.001	<0.001	Co-occurrence
MSK_Impact	173	4460	161	6151	0.360	0.014	0.043	Co-occurrence
Non-Redundant	203	4766	143	12,865	0.927	<0.001	<0.001	Co-occurrence

**Table 3 cells-10-02116-t003:** Co-occurrence frequencies of *TP53* alteration and *IDH1* hotspot mutation or isolated alteration in human cancer.

Dataset	R132		Isolated		Fisher’s Exact *p*-Value
TCGA_PanCancer	264/480	55%	118/491	24%	<0.0001
MSK_Impact	117/260	45%	45/74	61%	0.0179
Non-Redundant	89/217	41%	1782/7819	23%	<0.0001
Total	470/957	49%	1945/8384	23%	<0.0001

**Table 4 cells-10-02116-t004:** Co-occurrence frequencies of specific *IDH1*-R132 mutation and *TP53* alteration in human cancer.

Dataset	R132H	R132C	R132G	R132S	R132L	R132I
TCGA_PanCancer	220/389 (57%)	21/61 (34%)	13/16 (81%)	9/11 (82%)	1/3 (33%)	
MSK_Impact	90/168 (54%)	17/70 (24%)	5/9 (56%)	3/4 (75%)	2/8 (25%)	0/1 (0%)
Non-Redundant	74/115 (64%)	10/77 (13%)	5/9 (56%)	0/9 (0%)	0/7 (0%)	
Total	384/672 (57%)	48/208 (23%)	23/34 (68%)	12/24 (50%)	3/18 (17%)	0/1 (0%)

**Table 5 cells-10-02116-t005:** Co-occurrence frequencies of specific *IDH1* hotspot mutation and *TP53* alteration in different histological subtypes of glioma.

Cancer Type	TCGA_PanCancer		MSK_Impact		Combined	
	R132H	R132X	R132H	R132X	R132H	R132X
Astrocytoma	100/112 (89%)	19/21 (90%)	56/57 (98%)	9/9 (100%)	156/169 (92%)	28/30 (93%)
Glioblastoma	21/22 (95%)	3/3 (100%)	16/18 (89%)	3/3 (100%)	37/40 (93%)	6/6 (100%)
Oligoastrocytoma	64/98 (65%)	10/11 (91%)	11/16 (69%)	1/2 (50%)	75/114 (66%)	11/12 (92%)
Oligodendroglioma	33/147 (22%)	5/5 (100%)	3/67 (4%)	NA	36/214 (17%)	5/5 (100%)
Total	218/379 (58%)	37/40 (93%)	86/158 (54%)	13/14 (93%)	304/537 (57%)	50/54 (93%)

Co-occurrence of specific *IDH1* mutation and *TP53* alteration is expressed as a percentage of total count in each histological subtype of glioma.

**Table 6 cells-10-02116-t006:** Multivariate Cox proportional hazards analysis of *TP53* status in the *IDH1* hotspot mutation subgroup of TCGA-LGG dataset.

	Hazards Ratio	95% CI		*p*-Value
*TP53* no alteration vs. altered	2.079	1.083	3.992	0.028
*IDH1* R132H vs. R132X	0.348	0.122	0.991	0.048
Age <40 vs. >60 years old	4.649	2.25	9.608	<0.001
Male vs. Female	1.048	0.644	1.705	0.85
Oligodendroglioma vs. Astrocytoma	1.085	0.546	2.158	0.815

**Table 7 cells-10-02116-t007:** Multivariate Cox proportional hazards analysis of *TP53* status in the oligodendroglioma subgroup of TCGA-LGG dataset.

	Hazards Ratio	95% CI		*p* Value
*TP53* no alteration vs. altered	2.001	1.032	3.879	0.040
*IDH1* wildtype vs. R132	0.692	0.485	0.986	0.042
Age <40 vs. >60 years old	11.696	4.409	31.026	<0.001
Male vs. Female	0.787	0.401	1.545	0.486

## Data Availability

Available upon request.

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
