# Peer review of "Association of *TP53* Alteration with Tissue Specificity and Patient Outcome of *IDH1*-Mutant Glioma"

_cells, 2021, doi:10.3390/cells10082116_

Round 1

Reviewer 1 Report

In this paper, the authors focus on two key cancer genes, IDH1 and TP53, by performing a bioinformatic analysis of three datasets spanning 45228 human pan-cancer samples. In this data, they find that the prevalence of IDH1 mutants (hotspot mutations at R132) is overall rare (~3%) except in grade II/III gliomas, where they appear very common (77% in LGG data). The authors further find that in gliomas, IDH1 mutations are frequently associated with TP53 mutations (49%) and that most of these co-alterations occur specifically in lower-grade glioma and glioblastoma (97%). As the authors note, this frequent and specific association indicate a tissue-specific role in gliomagenesis, which the authors postulate could be related to ferroptosis.

The paper is well-written and interesting, and the conclusions appear warranted by the data. The IDH1 association to gliomas was previously known (see for example Zheng et al, 2019, Mol Cancer Ther), as is the lower prevalence of IDH1 mutations across other types of cancers (~1%, see for example Shen et al, 2021, MGGM); furthermore, the common co-occurrence of IDH1 and TP53 mutations in gliomas has been previously reported in several studies—which should therefore be cited in the manuscript—such as (Wang et al, 2014, Biomed Res Int; Mellai et al, 2011 J Neurooncol; Mukasa et al, 2012 Cancer Sci), in TERT-wt II/III gliomas (Yang et al, 2016, Neuro Oncol), and in relation with ATRX and BRAF mutations (Da et al, 2021, Front Oncol). However, while the association itself of IDH1 and TP53 was previously known, its specificity to gliomas is new, and in accordance with this both IDH1 mutation and TP53 status were found to be independent predictors of patient survival in glioblastomas (Wang et al, 2014, Asian Pac J Cancer Prev).

It would be interesting if the authors could investigate the status of several genes known to occur in gliomas in the 4 classes defined by IDH1 and TP53 status—such as TERT, ATRX, BRAF, or H3K27M (whose prognosis is also predicted by TP53 status, cf. Dong et al, 2018, Neurol Sci). This could detect with IDH1/TP53 gliomas are preferentially associated with other specific mutations.

Author Response

The IDH1 association to gliomas was previously known (see for example Zheng et al, 2019, Mol Cancer Ther), as is the lower prevalence of IDH1 mutations across other types of cancers (~1%, see for example Shen et al, 2021, MGGM); furthermore, the common co-occurrence of IDH1 and TP53 mutations in gliomas has been previously reported in several studies—which should therefore be cited in the manuscript—such as (Wang et al, 2014, Biomed Res Int; Mellai et al, 2011 J Neurooncol; Mukasa et al, 2012 Cancer Sci), in TERT-wt II/III gliomas (Yang et al, 2016, Neuro Oncol), and in relation with ATRX and BRAF mutations (Da et al, 2021, Front Oncol). However, while the association itself of IDH1 and TP53 was previously known, its specificity to gliomas is new, and in accordance with this both IDH1 mutation and TP53 status were found to be independent predictors of patient survival in glioblastomas (Wang et al, 2014, Asian Pac J Cancer Prev).

Response

We appreciate the reviewer’s comments on the relevant publications that we had overlooked in the previous submission. We have revised the manuscript accordingly by including relevant citations as follows:

Line 278: [37] (Shen et al, 2021, MGGM)

Line 279: [38] (Zheng et al, 2019, Mol Cancer Ther)

Line 328-32: [58,59] (Mukasa et al, 2012 Cancer Sci; Wang et al, 2014, Biomed Res Int)

Line 341-3: [61,62] (Yang et al, 2016, Neuro Oncol; Da et al, 2021, Front Oncol)

            Two of the suggested publications (Mellai et al, 2011 J Neurooncol and Wang et al, 2014, Asian Pac J Cancer Prev) were not cited because we respectively disagree that they are directly related to this study.  

“It would be interesting if the authors could investigate the status of several genes known to occur in gliomas in the 4 classes defined by IDH1 and TP53 status—such as TERT, ATRX, BRAF, or H3K27M (whose prognosis is also predicted by TP53 status, cf. Dong et al, 2018, Neurol Sci). This could detect with IDH1/TP53 gliomas are preferentially associated with other specific mutations.”   

Response

We appreciate this interesting suggestion. We have added this point in the discussion (Line 341-3) and the citation in Line 332-3 [60].

Reviewer 2 Report

This is a bioinformatic analysis of more than 45.000 human cancers that shows important information regarding the biology of IDH-mutant tumors such as gliomas. 

The information very well presented, methods are appropriate and the data shown are important for further studies but he study does not provide a clear interpretation of the findings regarding the association of TP53 alterations and IDH mutations in gliomas. Despite this fact, I believe it is important to publish this manuscript since it will be helpful for future projects on neuro-oncology. 

Author Response

None.